# Flexural Behavior of Corroded Concrete Beams Strengthened with Carbon Fiber-Reinforced Polymer

**DOI:** 10.3390/ma16124355

**Published:** 2023-06-13

**Authors:** Yiyuan Wang, Jin Wu

**Affiliations:** Department of Civil and Airport Engineering, Nanjing University of Aeronautics and Astronautics, 29 Jiangjun Road, Nanjing 211106, China

**Keywords:** CFRP, strengthening, corroded reinforced concrete beams, flexural capacity

## Abstract

The present study investigates the flexural behavior of reinforced concrete (RC) beams; the longitudinal reinforcing rebars of the beams were corroded and then strengthened with carbon fiber-reinforced polymer (CFRP). The corrosion of the longitudinal tension reinforcing rebars in eleven beam specimens was accelerated in order to obtain different corrosion levels. Afterwards, the beam specimens were strengthened by bonding one layer of CFRP sheets to the tension side to restore the strength loss due to the corrosion. The failure modes, flexural capacity, and midspan deflection of the specimens with different corrosion levels of the longitudinal tension reinforcing rebars were obtained by the four-point bending test. It was found that the flexural capacity of the beam specimens decreased with the increase in the corrosion level of the longitudinal tension reinforcing rebars and that the relative flexural strength was only 52.5% when the corrosion level was 25.6%. The stiffness of the beam specimens decreased significantly when the corrosion level was higher than 20%. Through a regression analysis of the test results, a model for the flexural bearing capacity of the corroded RC beams strengthened with CFRP was proposed in the study.

## 1. Introduction

Concrete structures are widely used in civil and architectural engineering. It is generally known that the major durability problem for RC structures under severe environments is the corrosion of the reinforcing rebars [1]. Due to the corrosion of reinforcing steel, many structures were weakened in terms of serviceability and safety and thus had to be repaired or strengthened.

In terms of structural performance, the corrosion of steel bars in concrete has a significant effect on the strength and stiffness of the structures [2]. The corrosion of reinforcing rebars results in a reduction in their cross-sectional areas, as well as a decrease in their strength, ductility, and other mechanical properties [3]. Corrosion products can generate expansive forces that alter the stress state in concrete and result in cracks and damage to beams. In addition, the corrosion of reinforcing rebars can also weaken the bond strength between the corroded rebars and the surrounding concrete. In particular, after the formation of cracks in concrete the ability of concrete to restrain reinforcing bars is reduced. Furthermore, the corrosion of reinforcing rebars can weaken the mechanical interaction between the deformed ribs of the rebars and the concrete. The aforementioned factors can have a significant impact on the flexural capacity and failure mode of the RC beams that have undergone corrosion [4,5,6].

It is necessary for the RC structures with corroded reinforcing rebars to be repaired or strengthened to restore the bearing capacity of the structure. The widely used strengthening methods for RC members include section enlargement, steel bonding, the external steel clad method, and CFRP strengthening. FRP composites have been successfully used to improve the strength and ductility of reinforced concrete members that have been severely damaged under different scenarios, such as earthquakes, fire, etc. [7]. Van Cao et al. [8] investigated the behavior of fire-exposed reinforced concrete slabs with FRP retrofitting. With the advantages of being lightweight and having convenient construction and good durability, it is an ideal method to strengthen the corroded reinforced concrete beam with a carbon fiber-reinforced polymer (CFRP) [9]. Therefore, many researchers focus on the flexural performance of corroded RC beams strengthened with CFRP.

Since the beginning of the 21st century, CFRPs have been more and more widely used in the rehabilitation of corroded RC structures. A pilot study carried out by Soudki and Sherwood [10] investigated the feasibility of bending corroded RC beams strengthened with CFRPs. They showed that CFRP sheets can restore the integrity and improve the capacity of the specimens; ten beams with three different corrosion levels (5%, 10%, and 15%), including six beams wrapped with CFRP sheets, were tested. The results showed the CFRP sheets can effectively restore the capacity of corrosion damaged concrete beams and increase the stiffness of the beams. Wang et al. [11] carried out an experimental and analytical program which consisted of twenty-four 200 mm × 350 mm × 3500 mm beams. The findings demonstrated that the strength and failure mechanisms of the retrofitted RC beams could be influenced by several key factors, including the level of corrosion in the reinforcing rebars, the water–cement ratio of the concrete, and the arrangement and number of the FRP patches. Kutarba [12] constructed thirty RC specimens to study the use of CFRP sheet repair as a rehabilitation technique for corroded RC beams. Following the repair and strengthening of twenty-six corroded beams using three distinct CFRP schemes, eight beams were load tested; at the same time, the remaining beams were subjected to the post-repair corrosion. The results showed that the application of CFRP significantly restored the bearing capacity of the specimens and provided a protective barrier capacity for the system to reduce the rate of secondary corrosion activity. Maaddawy et al. [13,14] designed an experiment which included eleven corroded beams; six of them were repaired with CFRP laminates, which extended the service life of the corroded RC beams. The research presented a new mathematical model for predicting the inelastic flexural response of corroded RC beams repaired with FRP laminates. The results showed that CFRP repair increased the ultimate strengths of the corroded beams but significantly reduced the deflection capacity at all levels of corrosion damage.

Many factors that affect the flexural performance of CFRP-strengthened corroded RC beams have been investigated by many scholars around the world. These factors include the corrosion of reinforcing rebars [15]; the mechanical behavior of the corroded reinforcement [16]; the quantity of CFRP sheets [17]; anchoring [18]; strengthening schemes [15]; the bond behavior of corroded reinforcing rebars in relation to the concrete interface [16]; initial load; and damaged concrete cover [19,20,21,22,23]. Kashi et al. [24] investigated the effect of marine environmental conditions on the durability of RC corroded columns strengthened with FRP sheets. Some researchers also investigated the structure performance of AFRP-strengthened corroded RC beams [25,26]. Gotame. M et al. [27] investigated the non-linear finite element analyses that had been conducted to assess the flexural behavior of corrosion-damaged RC beams strengthened with externally bonded FRP. Zheng, A et al. [28] studied the shear behavior of reinforced concrete (RC) beams with corrosion-damaged stirrups strengthened using a fiber-reinforced polymer (FRP) and a grid-reinforced engineered cementitious composite (ECC) matrix. Yang J. et al. [29] investigated the feasibility of using externally bonded FRP laminates combined with U-jackets, applied directly and without repairing the deteriorated concrete cover, to strengthen beams with corroded reinforcing rebars. The shear span-to-depth ratio can significantly influence the behavior of an RC beam shear strengthened with EBR–FRP composites and can even determine the shear failure mode of RC beams [30]. Chen et al. [31] proposed ultra-high-performance concrete (UHPC) combined with fiber-reinforced polymer (FRP) composites for the shear strengthening of corroded reinforced concrete beams.

In summary, domestic and foreign scholars have conducted many studies on the flexural behavior of corroded concrete beams strengthened with CFRP sheets. However, very limited information is available on the calculation method for the flexural capacity of corroded concrete beams strengthened with CFRP sheets. The present study reports the experimental findings on the flexural behavior of RC beams that underwent corrosion and were subsequently strengthened with CFRP sheets. The experimental study included eleven RC beams that were subjected to accelerated corrosion of the longitudinal reinforcing rebars at various levels. Following the corrosion, all the beams were strengthened by the bonding of one layer of CFRP sheets to the tension side to restore the strength loss due to the corrosion. Based on the regression analysis of the test results, a calculation approach was introduced for determining the flexural bearing capacity of the corroded RC beams strengthened with CFRP in this study.

## 2. Experimental Methods

### 2.1. Designed Corrosion Level and Strengthening Method

The designed corrosion level of the reinforcing rebars and the strengthening method used in the experiment are shown in Table 1. Accelerated corrosion was applied to all specimens, with the exception of the control beam (B0). Direct current (DC) power supply used in the accelerated corrosion was manufactured by Shanghai Wenkai Power Supply Equipment Ltd. in Shanghai, China. In the study, the corrosion level was defined as the mass loss of the reinforcing rebar. Eleven different corrosion levels of the longitudinal reinforcing rebars were designed, and all the specimens were strengthened with one layer of CFRP sheets.

### 2.2. Casting and Curing of Concrete Beams

This study employed river sand with a maximum size of 25 mm as the fine aggregate and coarse aggregate in the concrete mixture. The performances of the cement, fine aggregate, and coarse aggregate are shown in Table 2, Table 3 and Table 4. For all the specimens, the water–cement ratio of the concrete mixture was 0.4, whereas the masses of cement, water, coarse aggregate, and sand per cubic meter of concrete were 423.91 kg, 195 kg, 1148.8 kg, and 606.2 kg, respectively. The cubic concrete compressive strength was 30.28 MPa, as measured by pressure machine from Jinnan Shijin Group Ltd. in Jinan of China on the first day of structural testing. The longitudinal reinforcing rebars were deformed bars with the measured yield strength of 468.79 MPa and the ultimate strength of 628.45 MPa. The stirrups consisted of plain round bars with a diameter of 8 mm and a measured yield strength of 441.46 MPa. The CFRP composites were composed of dry fiber sheets and epoxy resin, and the measurements of the mechanical properties shown in Table 5 were given by the supplier of the carbon fibers and resin. The typical thickness of a single-layer composite CFRP sheet was 0.111 mm.

A cross-sectional view of the test specimen is presented in Figure 1, which provides details of its configuration. The test specimens were reinforced concrete beams, 1500 mm long and 120 mm wide with a 200 mm depth and 172 mm effective depth. Deformed steel rebars with a 10 mm diameter were used as tensile reinforcing rebars. The construction of the specimen was designed to produce flexural failure before shear failure occurred. To prevent anchorage failure, the tensile reinforcing rebars were equipped with a 90-degree hook at each end and extended 50 mm to attach the wires of the electrodes. A beam length of 1500 mm was selected, with a 1200 mm span between the supports. The span-to-effective depth ratio was 2.42, which enabled the shear span to be adequately reinforced with stirrups and to avoid shear failure. The mixing water in the concrete was enhanced with NaCl, added at a rate of 5% by weight of the cement, to expedite the corrosion. At the intersections of the stirrups, insulating tape and epoxy resin were put on the stirrups and the longitudinal reinforcing rebars were coated with insulating rubber to shield them from the corrosion.

### 2.3. Accelerated Galvanic Corrosion Process

To induce corrosion damage in the test specimens within a practical duration, an accelerated corrosion using direct current (DC) power supply manufactured by Shanghai Wenkai Power Supply Equipment Ltd. in Shanghai of China was employed. Following the curing process of the test specimens, the test beams were put in a watertight tank with 5% NaCl solution; the liquid level was above the surface of the beam specimens. The longitudinal reinforcing rebars served as the anode by being joined to the positive terminal of an external power supply, while a stainless steel rod was linked to the negative terminal of the power supply to function as a cathode.

Figure 2 illustrates the configuration employed for the accelerated corrosion process. To achieve a stable current density, a direct current (DC) power supply manufactured by Shanghai Wenkai Power Supply Equipment Ltd. in Shanghai of China was utilized, and a current density of 1 mA/cm^2^ was chosen. In accordance with Faraday′s law, the corrosion levels of reinforcing rebars were controlled by the time and amount of current passing through reinforcing rebars. The duration required to generate the corrosion impairment ranged from 27.16 h, which resulted in minor damage (B1), to 271.67 h, which produced severe damage (B10). Figure 3 shows the cracking of beam specimens after the corrosion of the reinforcing rebars.

### 2.4. Strengthening Regimes

All eleven specimens were strengthened with CFRP sheets and used the same strengthening scheme. Figure 4 displays the arrangement of the CFRP reinforcement. One layer of CFRP sheets with a 100 mm width was bonded to the tension face of the beam with a length of over 1500 mm and oriented in the longitudinal direction. 

Both ends of the beam were wrapped with two one-layer U-shaped CFRP sheets with the fibers oriented perpendicularly to the longitudinal axis. Each U-shaped CFRP sheet had a width of 100 mm, encompassing the tension face and extending upwards on either side to the top.

The surface of concrete zone to be bonded with the CFRP sheets was polished by a grinding machine manufactured by Changzhou Fangjia Electrical Company in Changzhou of China, and the corners of the beam specimens were chamfered to a radius of 20 mm to prevent stress concentration. After the surface was washed with alcohol, the CFRP sheets were bonded on the beam specimens with epoxy resin. Figure 5 shows the beam specimens after the rehabilitation with the FRP.

### 2.5. Test Setup and Instrumentation

After strengthening, the specimens needed to be cured for 7 days. To measure the concrete strain at the midspan depth of the beams, seven electrical strain gauges were employed. Additionally, electrical strain gauges were affixed to the longitudinal reinforcing rebars and the CFRP sheets. Figure 6 illustrates the configuration of the strain gauges.

Subsequently, all the beams were subjected to four-point bending tests until failure, with the loading points positioned 400 mm apart. The loading was gauged using two 200 kN pressure transducers via a hydraulic jack until failure occurred. The displacement of the beam at the midspan, loading point, and supports was monitored by installing five linear variable displacement transducers (LVDTs) (Donghua Testing Ltd., Taizhou, China) with a range capacity of 100 mm. At every load increment, the crack widths were measured with a visual crack comparator. 

## 3. Results and Discussion

### 3.1. Corrosion Damage and Inspection

After the accelerated corrosion of the longitudinal reinforcing rebars, the cracks due to the corrosion were mapped and recorded. The observed corrosion cracking patterns were consistent for all the corroded specimens. The distribution of cracks was non-uniform along the longitudinal reinforcing rebars due to the randomness of the occurrence of corrosion. The largest widths of the corrosion-induced cracks are shown in Table 6. 

After the loading failure of the beam specimens, three steel coupons were extracted from the corroded longitudinal reinforcing rebars. The rust was washed away by dilute hydrochloric acid; then, the reinforcing rebars were weighed, and the length was measured. Their per-unit-length weight loss was compared to those of the uncorroded coupons; then, the measured corrosion level was obtained. The designed and measured corrosion level of the tension reinforcing rebars is shown in Table 6.

### 3.2. Cracking Development and Failure Mode

After all the test equipment was checked, the loading test could be carried out. When the load was added to 13~15 kN, the first crack appeared in the tension section of the pure bending section of the beam specimens. As the load was increased, an increasing number of cracks emerged within the tensile zone. With the full appearance of the cracks, the width of the main cracks also increased with the increase in the load. 

When the load was close to the ultimate load, the CFRP sheets would wrinkle, delaminate, and tear, accompanied by a “crackling” sound that indicated the failure of the CFRP sheets. Eventually, the cracks extended to the compression zone of the beam; the compression zone concrete was crushed, and the CFRP sheets were debonded from the beam and ruptured. 

Crack profiles of beam specimens at ultimate loading are shown in Figure 7. The typical failure modes of the beam specimens are shown in Figure 8. The failure modes of all the specimens were flexible failures. The corrosion levels of the reinforcing rebars had little effect on the failure modes of all the beam specimens. 

### 3.3. Midspan Deflection Response

Figure 9 depicts the load–midspan deflection curves of all the beams. The load–midspan deflection curves of B0 are roughly divided into three stages by the inflection points. In the initial stage, the beam is within the elastic state prior to the emergence of the first cracks. After the first inflection point, the curve enters the second stage and the slope of the curve decreases slightly. After the second inflection point, the curve enters the third stage. The deflection of the beam specimens increase rapidly, and the curve changes and becomes gentle up to the peak load. In comparison to the control beam (B0), the stiffness of the corroded beam specimens was decreased due to the corrosion of the reinforcing rebars. 

As demonstrated in Figure 9, with the increase in the corrosion level, the bond behavior between the reinforcing rebar and the concrete was deteriorated, and the mechanism of the beam gradually transferred from “beam action” into “arch action”. With the increase in the corrosion level, the load–midspan deflection curves of all the corroded beams tended to be smoother, and the inflection point became less obvious due to the yield platform of the reinforcing bars becoming unapparent. In addition, the ultimate deflection of the beam specimens decreased significantly when the corrosion level was higher than 20%.

### 3.4. Strain Response of Tensile Steel Bar and CFRP

Figure 10 shows the load–strain relationships of the concrete, tensile reinforcing rebar, and CFRP sheet at the midspan of the beams. In Figure 10, it can be seen that the strain of the compressive concrete, tensile steel bar, and CFRP sheet increased continuously with the increase in the load. After the reinforcing bars yielded, the strain of the CFRP sheet increased fast. Finally, the CFRP ruptured after slightly debonding with the concrete. 

Figure 11 shows the strain of the CFRP at the midspan of the beam specimens; it can be seen that before cracking the load–CFRP strain curves of all the beam specimens are similar and almost linear with the increase in loading, but the strains of the CFRP increase suddenly at the cracking of concrete, indicating that CFRP undertakes more loads. 

The slopes of the load–CFRP strain curves decrease gradually with the increase in corrosion level of the reinforcing rebars due to the damage to the stiffness of the concrete beam cross-section. After the loads reach a certain value (named the critical point), the CFRP strains increase significantly until the ultimate strains, and the critical points are obvious for the low corrosion levels. With the increase in the corrosion levels, the critical points are not obvious and the load–CFRP strain curves become gradually smooth. 

Figure 12 shows the load–strain relationships of the CFRP at the anchored zone (C1), loading point C2 (C5) and midspan C3 (C4). It can be seen that the load–strain curve relationships of the CFRP at the different positions are different. The strains of CFRP at the anchored zone were smaller than the strains at the other positions, which proves that the anchorage was effective. At first, the strain of CFRP at the loading point and midspan increased quasi-linearly. However, after the yield of the tensile reinforcing rebars, the strains of CFRP at the midspan were faster and larger than the strain of CFRP at the loading point. This phenomenon was more and more obvious with the increase in the corrosion levels of the reinforcing rebar. However, when the CFRP sheets were nearly debonded, the strain of CFRP at loading point increased rapidly.

Table 7 shows the ultimate strain of the midspan of the CFRP sheets for all the specimens. The volume expansion of the rust production of the reinforcement rebars resulted in the cracking and spalling of the concrete cover, which may have led to the tensile strains of the CFRP sheets in beams B8, B9 and B10 being smaller than those of the rest of the test beams. The effective tensile strain of the CFRP was above 9000 με.

### 3.5. Cross-Section Analysis of The Beams

The strain distribution profiles over the depth of the midspan sections of the beam specimens are shown in Figure 13. It can be seen that the strains in the midspan sections of the beam specimens with and without CFRP were in good accordance with the plane section assumption.

### 3.6. Flexural Capacity

Table 7 presents the flexural ultimate capacity attained during testing. The relative flexural strength can be expressed as Equation (1).
(1)$$\eta =M_{c}/M_{0}$$
where *η* is the relative flexural strength; *M*_0_ is the flexural ultimate load of the uncorroded specimen; and *M*_c_ is the flexural ultimate capacity of the corroded specimens.

Figure 14 displays the correlation between the corrosion level of the tension reinforcing bars and the relative flexural strength. It can be seen that the relative flexural strength of the beam specimens decreased almost linearly with the increase in the corrosion level when the corrosion level was smaller than 22%, but the relative flexural strength decreased significantly when the corrosion level was larger than 22%; in particular, the relative flexural strength was only 52.5% when the corrosion level was 25.6%. The reason for this is that both the mechanical property of the tension reinforcing rebar and the bonding behavior between the tension reinforcing rebar and the concrete deteriorated. When the corrosion level was very high, the pit rust of the reinforcing rebar occurred, which resulted in the significant decrease in the cross-area and strength of the reinforcing rebar.

### 3.7. Analysis of Flexural Capacity of Beams Strengthened with CFRP

In this paper, the flexural capacity of corroded beams strengthened with CFRP is studied. First, to calculate the flexural capacity of uncorroded concrete beams strengthened with CFRP, this paper adopts the calculation formula of the *Code for the design of strengthening concrete structures* (GB50367-2013) [32].
(2)$$M_{0}\leq \alpha_{1}f_{c}bx\left(h-\frac{x}{2}\right)+f_{y0}^{\prime }A_{s0}^{\prime }\left(h-a^{\prime }\right)-f_{y0}A_{s0}\left(h-h_{0}\right)$$
(3)$$\alpha_{1}f_{c0}bx=f_{y}A_{s0}+\psi_{f}f_{f}A_{fe}-f_{y0}^{\prime }A_{s0}^{\prime }$$
(4)$$\psi_{f}=\frac{\left(0.8\varepsilon_{cu}h/x\right)-\varepsilon_{cu}-\varepsilon_{f0}}{\varepsilon_{f}}$$
(5)x≥2a′
where *M*_0_ is the flexural capacity of uncorroded concrete beams strengthened with CFRP; *x* is the height of the concrete compression; *b* and *h* correspond to the width and depth of the beam, respectively; $f_{y0}^{\prime }$ is the tensile strength and compressive strength of the tensile and compressive bars; $A_{s0}$ and $A_{s0}^{\prime }$ are sectional areas of the tensile bars and compressive bars; $a^{\prime }$ is the distance from the resultant force points of the compressive bars to the edge of the compressive section; *h*_0_ is the effective depth of the tensile reinforcing rebars; *f*_f_ is the tensile strength of CFRP; *A*_fc_ is the effective sectional area of CFRP; *ψ*_f_ is the strength utilization coefficient, considering that the practical tensile strain of CFRP cannot reach its design value: ψ_f_ = 1.0 for *ψ*_f_ >1.0; ε_cu_ is the maximum pressure strain of the concrete: ε_cu_ = 0.0033; ε_f_ is the designed value of the strain of CFRP; ε_f0_ is the delayed strain of CFRP when considering the effect of the secondary load; if not, ε_f0_ = 0. 

Taking the flexural capacity of the uncorroded concrete beams strengthened with CFRP as the base and the corrosion level of the tensile reinforcing rebar as the independent variable, the function of the corrosion reduction coefficient is established (Figure 14). 

Based on the analysis of the optimal fitting to the experimental data, a calculation formula for the reduction coefficient η of the flexural capacity of the corroded concrete beams strengthened with CFRP is derived
(6)$$\eta =\{\begin{array}{c} -0.0103\rho +0.987\ldots \ldots \ldots \ldots \ldots \ldots \ldots \ldots \ldots .0\%<\rho \%\leq 22\%\\ -0.0614\rho +2.1041\ldots \ldots \ldots \ldots \ldots \ldots \ldots \ldots ..22\%<\rho \%\leq 25\% \end{array}$$
where ρ is the longitudinal reinforcing rebar corrosion level, %.

The equation used to compute the flexural capacity of the corroded concrete beams strengthened with CFRP is
(7)$$M_{c}=\eta M_{0}$$
where *M*_c_ is the flexural capacity of the corroded concrete beams strengthened with CFRP and *M*_0_ is the flexural capacity of the uncorroded concrete beams strengthened with CFRP. 

The comparisons between the analytical flexural capacity (*M*_c_) calculated using Equation (7) and the experiment flexural capacity (*M*_exp_) from other researchers are shown in Table 8 and Figure 15. The mean value of *M*_exp_/*M*_c_ was 0.99∼1.25, and the coefficient of variation was 0.015∼0.066. It is found that Equation (7) is reliable and conservative.

## 4. Conclusions

Eleven beam specimens were constructed, and their longitudinal reinforcing rebars were subjected to accelerated corrosion. After the corrosion of the reinforcing rebars, the beam specimens were strengthened with CFRP sheets. The mechanical properties of all the specimens were measured. A model to predict the flexural capacity of the corroded strengthened beams was proposed in the study. Based on the results of the experiment and analyses, the following conclusions were obtained.

For all the beam specimens, the flexural failure was observed. The corrosion levels of the reinforcing rebars had little effect on the failure mode of all the beam specimens. All the beam specimens failed because the CFRP sheets were debonded from the beam specimens and ruptured.The strain distributions over the depth of the midspan sections of the beam specimens with and without CFRP were in good accordance with the plane section assumption.The stiffness of the beam specimens slightly decreased with the increase in the corrosion of the reinforcing rebars. However, the ultimate deflection of the specimens decreased significantly when the corrosion level was higher than 20%.The relative flexural strength decreased significantly when the corrosion level was greater than 22%; in particular, the relative flexural strength was only 52.5% when the corrosion level was 25.6%.Based on the regression analysis of the test results, the model for the flexural bearing capacity of the corroded reinforced concrete beams strengthened with CFRP was proposed by introducing the coefficient of the bearing capacity reduction. This model was verified by the data in the literature.

## Figures and Tables

**Figure 1 materials-16-04355-f001:**
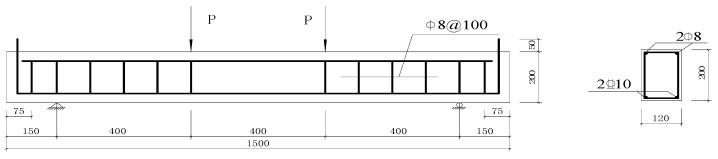
Details of test specimen.

**Figure 2 materials-16-04355-f002:**
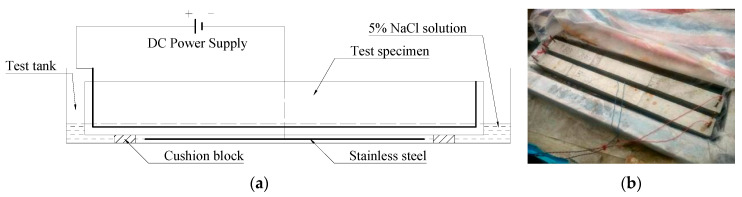
Setup for accelerated corrosion: (**a**) schematic of setup; (**b**) close-up view.

**Figure 3 materials-16-04355-f003:**
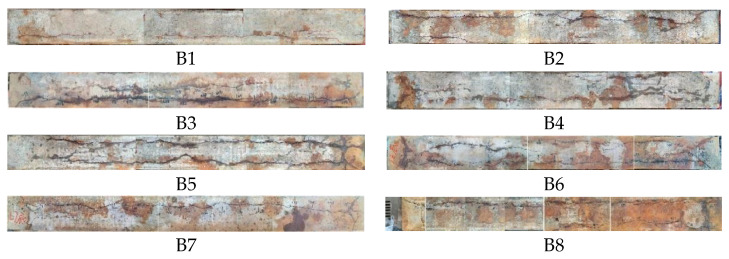
Photographs of bottom of beams after the corrosion.

**Figure 4 materials-16-04355-f004:**
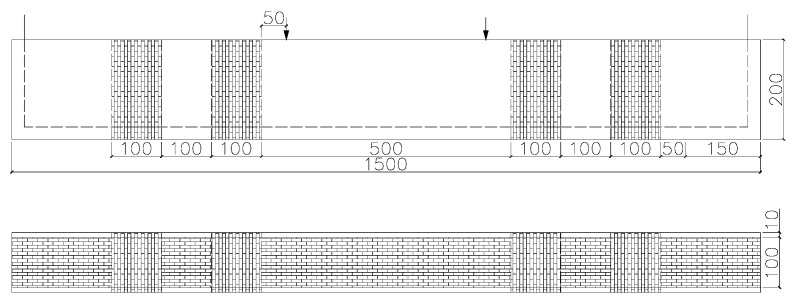
Details of strengthening methods.

**Figure 5 materials-16-04355-f005:**
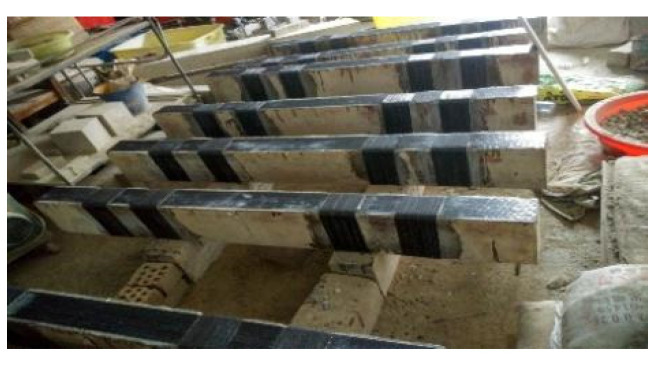
Photographs of beam specimens after the rehabilitation with the FRP.

**Figure 6 materials-16-04355-f006:**
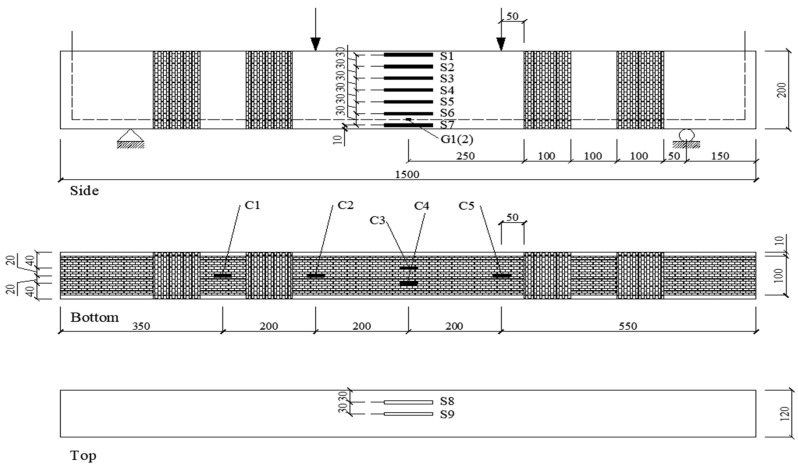
Schematic for gauges (all dimensions are in mm).

**Figure 7 materials-16-04355-f007:**
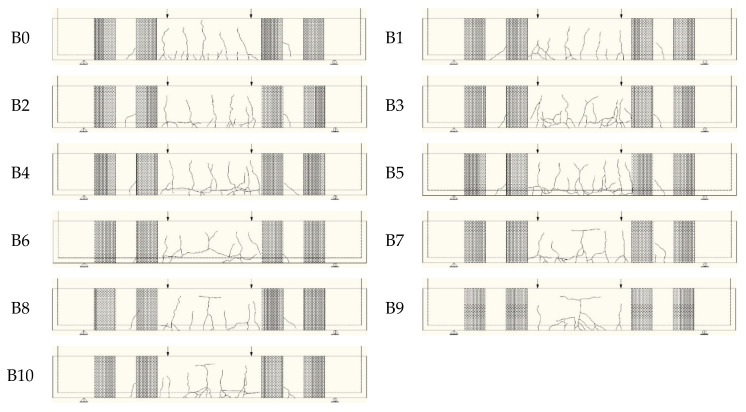
Crack profiles of specimens. (the arrow indicates the loading location).

**Figure 8 materials-16-04355-f008:**
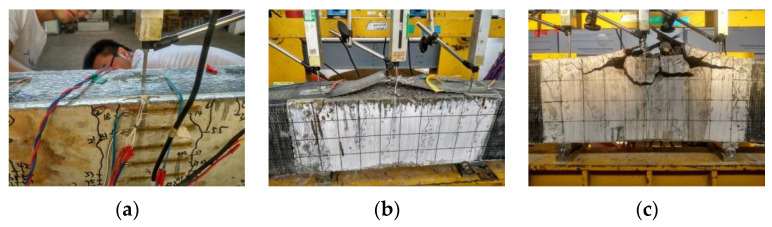
Typical failure modes: (**a**) CFRP debonding; (**b**) CFRP rupture; (**c**) concrete cover collapse.

**Figure 9 materials-16-04355-f009:**
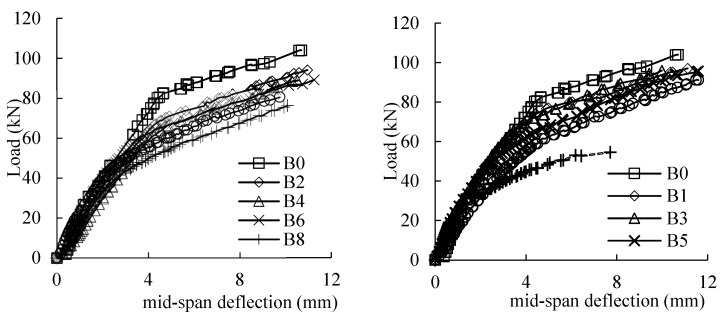
Load–deflection curves of specimens.

**Figure 10 materials-16-04355-f010:**
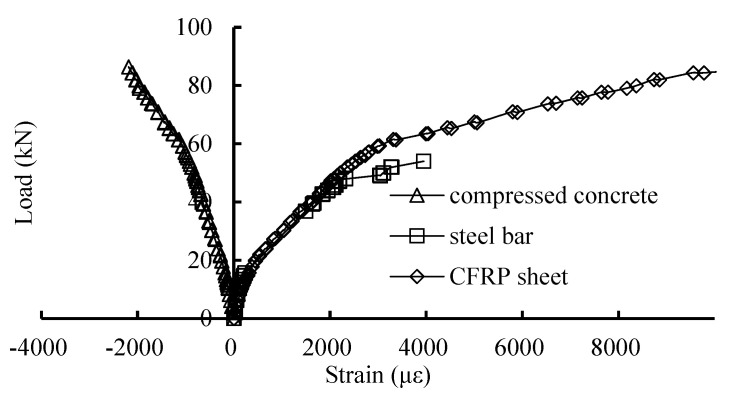
Longitudinal reinforcement and midspan fiber strain of beam B6.

**Figure 11 materials-16-04355-f011:**
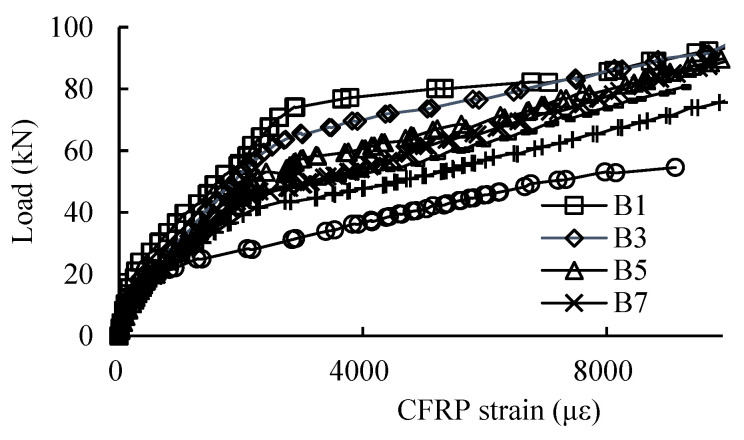
Load–strain curve relationships of CFRP at midspan of beam specimens.

**Figure 12 materials-16-04355-f012:**
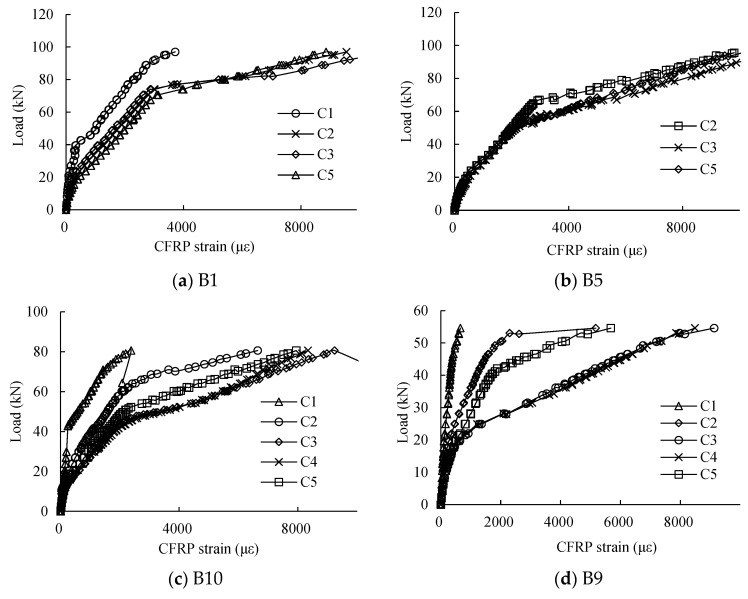
Load–strain curves relationships of CFRP at different positions of specimens.

**Figure 13 materials-16-04355-f013:**
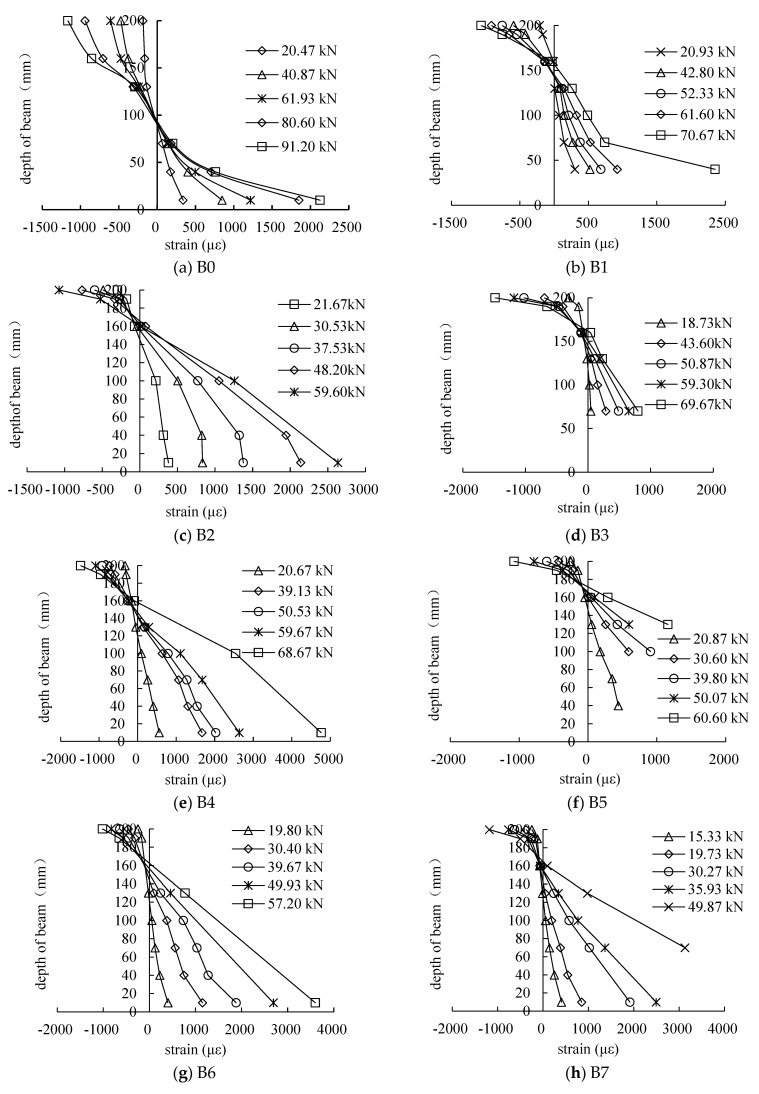
Strain distribution over the depth of midspan section of beam specimens.

**Figure 14 materials-16-04355-f014:**
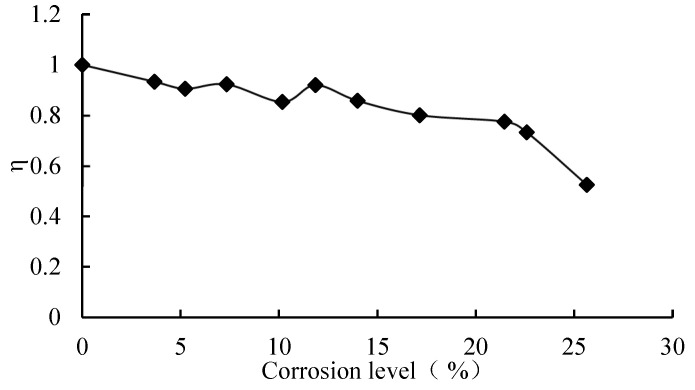
Relative flexural strength and corrosion level.

**Figure 15 materials-16-04355-f015:**
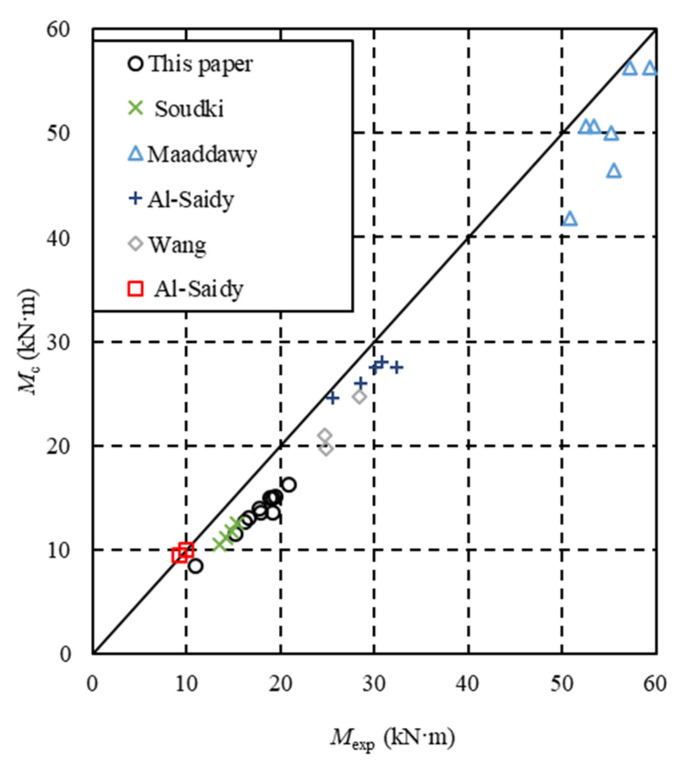
Comparison of *M*_exp_/*M*_c_ results from different researchers; Soudki [10], Maaddawy [13], Al-Saidy [15], Wang [16] and Al-Saidy [19].

**Table 1 materials-16-04355-t001:** Test matrix.

Specimen Number	Designed Corrosion Level (%)	Strengthening Method
B0	0	CFRP, one layer
B1	2	CFRP, one layer
B2	4	CFRP, one layer
B3	6	CFRP, one layer
B4	8	CFRP, one layer
B5	10	CFRP, one layer
B6	12	CFRP, one layer
B7	14	CFRP, one layer
B8	16	CFRP, one layer
B9	18	CFRP, one layer
B10	20	CFRP, one layer

**Table 2 materials-16-04355-t002:** Performance of cement.

Property	Value
Type and Class	P.C32.5
Specific surface area (m^2^ kg^−1^)	36.1
Initial and final setting times (min)	241/298
Three-day compressive and flexural strength (MPa)	17.4/3.3

**Table 3 materials-16-04355-t003:** Performance of fine aggregate.

Property	Value
Fineness modulus	2.91
Apparent density (kg m^−3^)	2560
Bulk density (kg m^−3^)	1245
Mud content (%)	0.2
Clod content (%)	0
Moisture content (%)	2.24
Grain grading	II

**Table 4 materials-16-04355-t004:** Performance of coarse aggregate.

Property	Value
Apparent density (kg m^−3^)	2583
Bulk density (kg m^−3^)	1296
Mud content (%)	0.3
Clod content (%)	0.6
Water absorption (%)	0.82
Elongated particle contents (%)	13
Crushing value (%)	12.5

**Table 5 materials-16-04355-t005:** Typical mechanical properties of dry carbon fibers and resin.

Material	Dry Carbon Fibers	Epoxy Resin
Tensile strength (MPa)	3060	35.5
Tensile modulus (MPa)	2.1 × 105	2040.6
Ultimate elongation (%)	1.6	2.1

**Table 6 materials-16-04355-t006:** Corrosion level of reinforcing rebars and cracks width due to corrosion.

Specimen Number	Corrosion Level (%)		Crack Width (mm)
	Designed	Measured	
B0	0	0	0
B1	2	3.66	0.2
B2	4	5.22	0.3
B3	6	7.33	0.5
B4	8	10.16	0.5
B5	10	11.85	0.4
B6	12	13.99	0.7
B7	14	17.15	0.4
B8	16	22.59	0.8
B9	18	25.64	1
B10	20	21.45	0.8

**Table 7 materials-16-04355-t007:** Flexural ultimate load and ultimate strain of midspan fiber.

Specimen Number	*ρ* (%)	*P*_u_ (kN)	Strain (με)
B0	0	104.00	-
B1	3.66	97.07	11,334
B2	5.22	94.13	11,031
B3	7.33	96.00	10,568
B4	10.16	88.73	10,336
B5	11.85	95.67	11,311
B6	13.99	89.20	11,194
B7	17.15	83.27	10,354
B8	22.59	76.27	9966
B9	25.64	54.60	9128
B10	21.45	80.60	9239

**Table 8 materials-16-04355-t008:** Comparison of *M*exp/*M*c results from different researchers.

Researchers	Specimen	*ρ*/%	*M*_exp_/kNm	*M*_c_/kNm	*M*_exp_/*M*_c_
Soudki [10]	CF-0	0	15.34	12.60	1.22
	CF-5	5	14.76	11.79	1.25
	CF-10	10	14.16	11.14	1.27
	CF-15	15	13.54	10.49	1.29
Maaddawy [13]	UR-1	0	57.17	56.29	1.02
	UR-2	0	59.40	56.29	1.06
	CRN-50-1	8.5	53.41	50.63	1.05
	CRN-50-2	8.5	52.54	50.63	1.04
	CRS-50-1	9.5	55.17	50.05	1.1
	CRS-110-1	15.7	55.46	46.46	1.19
	CRS-210-1	23.7	50.84	41.82	1.22
Al-Saidy [15]	M5S1	5	30.15	27.54	1.09
	M5S2	5	32.31	27.54	1.17
	M10S2	10	28.53	26.02	1.1
	M15S2	15	25.61	24.51	1.04
	M15S2-2L	15	30.83	27.96	1.1
Wang [16]	00-S2	0	28.44	24.67	1.15
	10-S2	10.65	24.72	20.95	1.18
	20-S2	16.65	24.78	19.74	1.26
Al-Saidy [19]	S5	5	8.53	10.61	0.8
	S7.5	7.5	9.75	10.32	0.95
	S10	10	8.92	10.02	0.89
	SP10	10	9.99	10.02	1
	SP15	15	9.17	9.44	0.97
	S15	15	8.29	9.44	0.88

## Data Availability

Not applicable.

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
