# Peer review of "Flexural Behavior of Corroded Concrete Beams Strengthened with Carbon Fiber-Reinforced Polymer"

_materials, 2023, doi:10.3390/ma16124355_

Round 1

Reviewer 1 Report

This research experiment is well carried and results and discussions are appropriately presented. I recommend this paper for publication after minor revisions.

Minor English corrections and editing is necessary before publication.

Author Response

Thank you for your comments very much. We have revised the manuscript according to your comments.

Reviewer 2 Report

The authors investigated the flexural behavior of corroded concrete beams strengthened  with carbon fiber reinforced polymer. They studied the effect of different corrosion levels on the main mechanical properties of he beams such as flexural failure, stiffness, and flexural capacity. The research work fits well with the scope of the journal, and the topic is interesting for scientific community. However, there are several work points in this work which should be solved before publishing this work. Here are some comments to improve the overall quality of the manuscript: 

1.       Abstract

·         Add some numerical results.

2.       Table 5

·         The composition of the different series is the same, so there is no need for this table. You can just mention that in the text.

3.       Table 6

·         Write the source of these measurements.

4.       Experimental methods

·         Refer to the standard tests such as ASTM

5.       Tables 7 & 8

·         Merge these two tables.

6.       Tables 9 and 10

7.       Merge these two tables

8.       References

9.       All the references are older than 5 years, please add recent references.

Author Response

Thank you for your comments very much and taking too much time on review of manuscript. We have revised the manuscript according to your comments.

Point 1:Abstract

  • Add some numerical results.

Response 1: We have added some numerical results in abstract.

Point 2. Table 5

The composition of the different series is the same, so there is no need for this table. You can just mention that in the text.

Response 2: We have deleted Table 5, and the composition is mentioned in the manuscript.

Point 3. Table 6

Write the source of these measurements.

Response 3: The source of measurements in Table 6 is the supplier of carbon fibers and resin, and is added in the manuscript.

Point 4. Experimental methods

Refer to the standard tests such as ASTM

Response 4: In this study, we refered to test standard of China, such as” AQSIQ (General Administration of Quality Supervision, Inspection and Quarantine of the People’s Republic of China). (2007). “Common Portland cements.” GB 175-2007, Standards Press of China, Beijing”, “AQSIQ (General Administration of Quality Supervision, Inspection and Quarantine of the People’s Republic of China). (2009). “Standard for test methods of long-term performance and durability of ordinary concrete.” GB/T 50082-2009, Standards Press of China, Beijing”.

These test standard of China are similar to ASTM.

Point.5. Tables 7 & 8

  • Merge these two tables.

Response 5: We have ·merged these two tables.

Point.6. Tables 9 and 10

Merge these two tables

Response 6: We have ·merged these two tables.

Point.7 References

All the references are older than 5 years, please add recent references.

Response 6: We added five references in the recent five years.

Reviewer 3 Report

My comments are following:

1. In last part of abstract section, more details of obtained results should be provide.

2. Introduction section should be improve with highlight the problem, gap of and novelty of this study.

3. Table 5 can be revise (B0-10) as all simlair content. 

4. Figs. 4 and 5 should be revise with more details.

5. References, section and sub-section should be follow the publisher style.

6. For the results, More discussion/details and compare to literature should be provide.

7. Authors should re-write the conclusion section with more details about obtained results. 

Authors should revise the manuscript and improve the English writing. 

Author Response

Thank you for your comments very much and taking too much time on review of manuscript. We have revised the manuscript according to your comments.

Point 1: In last part of abstract section, more details of obtained results should be provide.

Response 1: We added some numerical results in last part of abstract as following.

Abstract: This paper presents the results of research on the flexural behavior of reinforced concrete beams, which longitudinal reinforcements were corroded, strengthened with carbon fiber reinforced polymer (CFRP). The corrosion of longitudinal tension reinforcements in 11 reinforced concrete beam specimens were accelerated to obtain different corrosion levels. After that all beam specimens were strengthened by bonding one layer CFRP sheet to the tension side to restore the strength loss due to corrosion. Finally, the failure modes, flexural capacity, mid-span deflection of the specimens with different longitudinal tension reinforcement corrosion levels were obtained by the four-point bending test. It was found that the flexural capacity of beam specimens decreased with the increase of the longitudinal reinforcement corrosion level, and the relative flexural strength was only 52.5% when the corrosion level was 25.6%. The stiffness of beam specimens decreased significantly with the increase of corrosion level when the corrosion level was larger than 20%. Based on the regression analysis of the test results, a calculation method for flexural bearing capacity of reinforced concrete beams with corroded longitudinal reinforcement strengthened with CFRP was proposed.

Point 2: Introduction section should be improve with highlight the problem, gap of and novelty of this study.

Response 2: We revised the introduction section as following.

In summary, domestic and abroad scholars have done many studies in flexural behavior of corroded concrete beams strengthened with CFRP sheets. However, very limited information is available on the calculation method of flexural capacity of corroded concrete beams strengthened with CFRP sheets. This paper presents results of an experimental study designed to investigate on the flexural behavior of reinforced concrete beams with corroded longitudinal reinforcement strengthened with CFRP sheets. The experimental program consisted of eleven reinforced concrete beams exposed to accelerated corrosion on different corrosion levels of longitudinal reinforcement. Then all the beams were strengthened by bonding one layer CFRP sheet to the tension side to restore the strength loss due to corrosion. Based on the regression analysis of the test results, a calculation method for flexural bearing capacity of reinforced concrete beams with corroded longitudinal reinforcement strengthened with CFRP was proposed in this study.

Point 3. Table 5 can be revise (B0-10) as all simlair content.

Response 3: We have deleted Table 5, and the composition is mentioned in the manuscript.

Point 4. Figs. 4 and 5 should be revise with more details.

Response 4: More details were added in the Fig. 4.

Fig. 5 was deleted due to no useful information.

Point 5. References, section and sub-section should be follow the publisher style.

Response 5: We revised the style of references, section and sub-section.

Point 6. For the results, More discussion/details and compare to literature should be provide.

Response 6: For the results, we added the relevant discussion.

Point 7. Authors should re-write the conclusion section with more details about obtained results.

Response 7:We have re-write the conclusion as following.

“Conclusions

Eleven beam specimens were constructed and their longitudinal reinforcing rebars were subjected to accelerated corrosion. After the corrosion of reinforcing rebars, beam specimens were strengthened with CFRP sheets. The mechanical properties of all the specimens were measured. The model to predict flexural capacity of corroded strengthened beams was proposed in the study. Based on the results of the experiment and analyses, the following conclusions are obtained.

  1. For all beam specimens, the flexural failure was observed. the corrosion levels of reinforcing rebars had little effect on the failure mode for all the beam specimens. All beam specimens failed because the CFRP sheets were debonding from the beam specimens and ruptured.
  2. The strain in the midspan section of beam specimens with and without CFRP were in good accordance with the plane-section assumption.
  3. The stiffness of beam specimens slightly decreased with the increase of the corrosion of reinforcing rebars. But the ultimate deflection of specimens decreased significantly when the corrosion level was larger than 20%.
  4. The relative flexural strength deceased significantly when the corrosion level was larger than 22%, especially the relative flexural strength was only 52.5% when the corrosion level was 25.6%.
  5. Based on the regression analysis of the test results, the model for flexural bearing capacity of corroded reinforced concrete beams strengthened with CFRP was proposed by introducing the coefficient of the bearing capacity reduction. This model was verified by the data in the literature.

Reviewer 4 Report

Paper:  materials-2377790

Title: Flexural behavior of corroded concrete beams strengthened 2 with carbon fiber reinforced polymer

The paper is interesting, well written and very useful for the practicing engineers.

Eleven (11) reinforced concrete beams have constructed and tested for the needs of the presented investigation. The specimens were exposed to accelerated corrosion and then all the beams were strengthened by bonding one layer of CFRP sheet on the tension side to restore the strength loss due to the corrosion.

- The introduction although informatory enough, could also include more supporting information about the successful use of the FRP sheets applied for the retrofitting of damaged reinforced concrete elements.

- Further, the well-known strengthening technique commonly used after structural damage is based on the application of FRP-sheets after the repairing of internal damage by infusing thin resin under pressure in the cracking system of the damaged body. The presented technique is based on the application of FRP-strips or sheets without prior repairing of internal damage. Recently, the effectiveness of the application of FRP sheets, after a superficial repair of the cracks without repairing the internal damage, for the rehabilitation of damaged RC beam-column joints (as in the examined cases), is experimentally investigated (take a look at the work “Full scale tests of RC joints with minor to moderate seismic damage repaired using C-FRP sheets”, Earthquakes and Structures, Vol. 15, No.6, pp. 617-627, 2018 and “Effectiveness of the novel rehabilitation method of seismically damaged rc joints using c-frp ropes and comparison with widely applied method using c-frp sheets—experimental investigation”, Sustainability (Switzerland) Vol. 13, Issue 11, 2021 Article number 6454).

- Cross section analysis of the beams (with and without CFRPs) can be performed and the results be mentioned in the revised manuscript.

- How do you define the corrosion levels (Table 1)? Add comments about the corrosion levels.

- It is noted that all lines of Table 5 are the same. Therefore, instead of Table 5 it could be stated that for all specimens the Water-binder ratio is 0.46 whereas cement, water, graved and sand are 423.91 , 195 , 1148.8 and 606.2 , respectively.

- More photographs of the cracking systems developed after the corrosion (along with figure 5) have to be added. Also, photographs after the rehabilitation with the FRPs, prior and after the testing, could help the reader to understand the efficiency of the applied technique.

- A thorough re-check of the English grammar is required throughout the manuscript.

Final conclusions

The submitted paper is a good experimental work. Further, the paper presents a good elaboration and evaluation of the test results and includes useful comments on the parameters. Amendments and improvements based on the aforementioned suggestions may help the authors to enhance their good work. A minor (but carefully conducted) revision is recommended (along with a re-check of the English grammar).

A thorough re-check of the English grammar is required throughout the manuscript.

Author Response

Thank you for your comments very much and taking too much time on review of manuscript. We have revised the manuscript according to your comments.

Point 1: The introduction although informatory enough, could also include more supporting information about the successful use of the FRP sheets applied for the retrofitting of damaged reinforced concrete elements.

Response 1: We added three references about the successful use of the FRP sheets applied for the retrofitting of damaged reinforced concrete elements.

  1. Maheswaran, M. Chellapandian N., Arunachelam.(2022). Retrofitting of severely damaged reinforced concrete members using fiber reinforced polymers: A comprehensive review. Structure. 38 (3):1257–1276

Amin Kashi, Ali Akbar Ramezanianpour, Faramarz Moodi(2017). Durability evaluation of retrofitted corroded reinforced concrete columns with FRP sheets in marine environmental conditions. Constr. Build. Mater., 151 (10):520–533

Vui Van Cao, Huy Ba Vo, Luan Hoai Dinh, et al (2022). Experimental behavior of fire-exposed reinforced concrete slabs without and with FRP retrofitting. J. Build. Eng. 51 (3):104315

Point 2:- Further, the well-known strengthening technique commonly used after structural damage is based on the application of FRP-sheets after the repairing of internal damage by infusing thin resin under pressure in the cracking system of the damaged body. The presented technique is based on the application of FRP-strips or sheets without prior repairing of internal damage. Recently, the effectiveness of the application of FRP sheets, after a superficial repair of the cracks without repairing the internal damage, for the rehabilitation of damaged RC beam-column joints (as in the examined cases), is experimentally investigated (take a look at the work “Full scale tests of RC joints with minor to moderate seismic damage repaired using C-FRP sheets”, Earthquakes and Structures, Vol. 15, No.6, pp. 617-627, 2018 and “Effectiveness of the novel rehabilitation method of seismically damaged rc joints using c-frp ropes and comparison with widely applied method using c-frp sheets—experimental investigation”, Sustainability (Switzerland) Vol. 13, Issue 11, 2021 Article number 6454).

Response 2: In this study, before bonding FRP sheets to beam specimens, the surface of specimens only was polished by a grinding machine and washed with alcohol but without prior repairing of corrosion cracks. In further work, beam specimens damaged due to corrosion will be strengthened with FRP-sheets after the repairing of corrosion cracks by infusing thin resin under pressure.

Point 3:- Cross section analysis of the beams (with and without CFRPs) can be performed and the results be mentioned in the revised manuscript.

Response 3: Cross section analysis of the beams with and without CFRP was added and the result(The strain in the midspan section of beam specimens with and without CFRP were in good accordance with the plane-section assumption)was mentioned in the revised manuscript.

Point 4:- How do you define the corrosion levels (Table 1)? Add comments about the corrosion levels.

Response 4: In this study, the corrosion level is defined as mass loss of reinforcing rebar.

  The definition of corrosion level was added in the manuscript.

Point 5:- It is noted that all lines of Table 5 are the same. Therefore, instead of Table 5 it could be stated that for all specimens the Water-binder ratio is 0.46 whereas cement, water, graved and sand are 423.91 , 195 , 1148.8 and 606.2 , respectively.

Response 5: Table 5 was deleted and the mixture was stated in the manuscript.

Point 6:-- More photographs of the cracking systems developed after the corrosion (along with figure 5) have to be added. Also, photographs after the rehabilitation with the FRPs, prior and after the testing, could help the reader to understand the efficiency of the applied technique.

Response 6: The photographs of the cracking of beam specimens after the corrosion and after the rehabilitation with the FRP have been added in the manuscript, as shown in Fig.3 and Fig.5.

Point 7:-- A thorough re-check of the English grammar is required throughout the manuscript.

Response 7:  The manuscript has been revised throughout.

Round 2

Reviewer 2 Report

The authors have addressed my remarks

Reviewer 4 Report

Revised Paper:  materials-2377790

Title: Flexural behavior of corroded concrete beams strengthened with carbon fiber reinforced polymer

As stated in the first review the paper is interesting well written and very useful for the practicing engineers.

In the revised manuscript the introduction has been improved.

Cross section analysis of the beams with and without CFRP was added in the revised in the manuscript.

Definition of the corrosion levels is defined as mass loss of reinforcing rebar.

Table 5 was deleted and the mixture was stated in the text.

Table

Final conclusions

Thus, the authors have responded rather successfully (more or less) to the reviewer’s comments. The submitted paper is a good experimental work.

Acceptance is recommended.

Minor editing.